# Sustainability in the European Union: Analyzing the Discourse of the European Green Deal

**Eva Eckert** [1,2,*] **and Oleksandra Kovalevska** [3,*]

1    School of Humanities & Social Sciences, Anglo-American University, Letenská 120/5, 118 00 Malá Strana, Czech Republic
2    Department of Linguistics, Charles University, Prague 1, 116 36 Prague, Czech Republic
3    Department of International Relations and European Studies, Metropolitan University Prague, Dubečská 900/10, 100 31 Strašnice, Czech Republic
*    Correspondence: eva.eckert@aauni.edu (E.E.); sasha.kovalevska1@gmail.com (O.K.)

**Abstract:** In the European Union, the concern for sustainability has been legitimized by its politically and ecologically motivated discourse disseminated through recent policies of the European Commission and the local as well as international media. In the article, we question the very meaning of sustainability and examine the *European Green Deal*, the major political document issued by the EC in 2019. The main question pursued in the study is whether expectations verbalized in the *Green Deal*'s plans, programs, strategies, and developments hold up to the scrutiny of critical discourse analysis. We compare the *Green Deal*'s treatment of sustainability to how sustainability is presented in environmental and social science scholarship and point out that research, on the one hand, and the politically motivated discourse, on the other, do not correlate and often actually contradict each other. We conclude that sustainability discourse and its keywords, lexicon, and phraseology have become a channel through which political institutions in the EU such as the European Commission sideline crucial environmental issues and endorse their own presence. The *Green Deal* discourse shapes political and institutional power of the Commission and the EU.

**Keywords:** sustainability; critical discourse analysis; the *European Green Deal*; plastic waste; economic growth

## 1. Introduction

The *European Green Deal* (The European Green Deal 2019) published by the European Commission (EC), the policy-issuing body of the EU, is the most recent set of policy initiatives on sustainable development that represents an opportunity for the EC to explain sustainability practices and prioritize the issue of environmental protection. At the same time, the *Green Deal* is the culmination of EU-wide discourse on the topic that has been built over the past four decades. During the current COVID-19 pandemic, environmental protection has become temporarily marginalized but the environment benefited from the decrease in greenhouse gases and air pollutant emissions caused by the reduction of economic activity and transport use (Pihl et al. 2021). Nevertheless, we are in the middle of an environmental catastrophe that must soon return to the headlines, captivating worldwide attention and demanding environmental action.

The EC has prioritized environmental issues in its development strategies. However, only the so-called critical discourse analysis (CDA) of how it "talks" about them can reveal the actual thinking and planning invested in the *Green Deal*, and unpack its content. These are the objectives tackled in the present paper that is framed by the CDA theory and that uses its method of inquiry (see Fairclough 1989 and 1995; Pennycook 2001; van Dijk 2008). We were inspired, in particular, by (Blommaert and Bulcaen 2003; Buckledee 2018; Kambites 2014; and Kress and Hodge 1979) acknowledging the fact that the very politics of language bring an agenda into fruition. (Cohen 2010;

Nichols 2007; and Stubbs 2001) clarified how the very word count can reveal what topics are overt and which ones are presented in a covert manner.

Scholars in political science and policy research see the need to unpack the concept of sustainable development and introduce diverse strategies to implement what the *Green Deal* laid out. For example, (Ossewaarde and Ossewaarde-Lowtoo 2020) question the supposedly revolutionary impact of the *Green Deal* on European future since the EC discourse has remained unchanged in its assessment and promotion of the so-called green growth empowering environmental protection. Siddi (2016) questions the ability of the EU to implement the strategies outlined in the *Green Deal*, which will depend on adequate financial resources, policy priorities, etc. A detailed linguistic analysis goes beyond the above-mentioned scholarly literature. Considering the prominence and novelty of the *Green Deal*, we build upon (van Dijk 2008; Fairclough 1989, 1995; and 2001; Fidler and Václav 2018), among others, to check for its overall content and agenda as well as consistency and fairness in treating crucial themes (in particular those that have been traditionally difficult to manage and represent) from the CDA perspective. To account for variation in the sustainability discourse and set the *Green Deal* in context of comparable documents prepared by the EC, we have reviewed the above-mentioned documents and critically analyzed *A European Strategy on Plastics in a Circular Economy* (2018). The sustainability agenda has been recently incorporated into research handbooks on finance, investment, and corporate management, aiming to encourage social and environmental objectives in the relevant fields (Boubaker et al. 2018; Boubaker and Nguyen 2019).

A major contribution of this paper lies in the assertion that the *Green Deal* sidelines a crucial issue, i.e., the practice of the so-called international waste trade in treating environmental protection. This issue is judged unilaterally as a harmful practice by contemporary scholarship that deals with growth vs. degrowth and technology in sustainability, yet, it is mostly excluded in the *Strategy on Plastics* and only briefly mentioned in the *Green Deal*, which we find surprising in a document addressing Europe's environmental future (cf. Song 2016; Ward et al. 2016). Admittedly, the EC cannot deal with all the issues at once but the European Environmental Agency (EEA) singled out waste trade alias plastic management as demanding urgent attention (European Environmental Agency 2019). The critical analysis and comparison of both texts has shown that the EU has legitimized not only beneficial but also harmful environmental practices in the documents and its overall discourse. The EC acts as the EU agent, leading Europe into an "eco-friendly" future and an expert with considerable symbolic power (cf. Bourdieu 1991). To maintain this status, it has highlighted positive developments and accomplishments but tended to sideline unpopular and controversial topics lacking clear solutions such as international waste trade, on the one hand, and growth vs. degrowth, on the other, and countering the ideology associated with the EU, i.e., that economic growth entails progress and better future.[1]

Given the current power structure and ideology, it will be exceedingly challenging for the EU to manage these topics, despite the supposedly revolutionary approach of the *Green Deal.* To support this suggestion, we analyze the actual discourse of the document and publicize the findings. Moreover, we compare treatment of sustainability in the *Green Deal* to how it has been presented in environmental and social science scholarship and point out that sustainability research, on the one hand, and the politically motivated discourse of sustainability, on the other, do not correlate and often actually contradict each other. We also investigate the ways in which the *Green Deal* discourse has shaped political and institutional power of the EU and speculate that its primary goal has been to advance the EU as the mover, expert, and leader in the field, seeking to generate public trust and hegemony by depending on the current presence, circulation, and ongoing innovation of the language it uses, i.e., the keywords, lexicon, grammatical patterns, and phraseology (see

---

[1] At the *Goals and values* section of the EU official website *sustainable development* is correlated with *balanced economic growth and price stability, a highly competitive market economy with full employment and social progress, and environmental protection* (https://europa.eu/european-union/about-eu/eu-in-brief_en).

Fairclough 1989 and 1995; van Dijk 2008). Our results show clearly that the EC opted for language of business, institutional diplomacy, economy, and technology to communicate the future of sustainability to the public. However, its sort of discourse and lexicon disable communication, which contradicts the intentions of the EC. The reasons are as follows: (1) The document underestimates the citizens' will to act; it may suggest inclusive transition but does not call for a reassessment of behavioral norms or an imminent social change; (2) although the EC's strategy rests in environmental transformation as a globally shared commitment, given the emphasis on power, it is unlikely for the *Green Deal* to engage the EU's citizens to the degree that they adopt the transformation for their own sake; (3) the power structure and ideology of the EU as the symbolic elite stand in the way of managing the very topics outlined in the *Green Deal*; (4) the linguistic analysis made it evident that concepts, keywords, set phrases, and grammatical patterns of the sustainability register endorsed the traditional agenda supporting the EU as the controlling institution; (5) technical difficulty of the lexicon, lack of clarity in transferring meanings, and ambiguity in identifying the very agents and actions further endorsed institutional rather than public ownership of the discourse; (6) and finally, the jargon of sustainability weakened the actual content. In sum, the authors suggest that the discourse reveals a continued tolerance of socio-economic and political power structures responsible for the ecological crisis that frames the *Green Deal* in a new way rather than endorsing a true change in thinking about ecology in the EU future.

In the follow-up sections, we describe the methodology, provide the context of the *Green Deal*, analyze its content, trace how the discourse manifests itself in various public domains, introduce the readers to an alternative discourse on sustainability, and conclude with outlining implications of the discourse.

## 2. Methodology

Critical discourse analysis is a research strategy to reveal sidelined agenda and undeclared goals, and to identify themes through which a topic is constructed as well as those it avoids. In our case study, it became a way to discover what sustainability implied when instituted as a political program by the EC. Our analysis was directed primarily at the language in the documents that we consider estranged from the daily reality of environmental protection (cf. Pennycook 2001). We examined the content of *The Green Deal for Europe* through its phraseology, concepts, keywords, and sentential patterns, applying the case study method and that of critical discourse analysis. Particular attention was paid to set phrases introduced to coin concepts dealing with sustainability, the registers feeding the content, as well as the style used to represent it. We identified and counted keywords along with their contexts, assuming that any text can be characterized in terms of its prominent linguistic units (cf. see also Stubbs 2001; Fidler and Václav 2018, 198). We looked into the distribution of grammatical structures and noted recurring verbs, noun phrases, patterns, and sentential constructions marking agents of particular actions or enabling agent deletion. Our linguistic inquiry is framed by studies of critical linguistics and discourse analysis (Cohen 2010; Fairclough 2001; Nichols 2007; van Dijk 2008) and has drawn upon findings in recent research on the discourse of sustainability (Jindřichovská et al. 2020; Machin 2019; Ossewaarde and Ossewaarde-Lowtoo 2020).

We also applied the Linguistic Analysis and Word Count (LIWC) 2015 software Macintosh v.1.6 to conduct the textual analysis (which included percentage-based count of grammatical patterns, clout, and other textual features) and word category analysis aiming at counting words associated with particular concepts such as power, money, et al. After running the text of the *Green Deal* through the program, we picked the most relevant results manually and excluded results we considered irrelevant.[2] LIWC enabled us to answer the question how the key topic of European future in relation to sustainability

---

[2] For example, two most used words in the adjective category were *neither* and *as* that we excluded in Results but did include the words following them, i.e., *affordable, free, rich* et al.

was presented and developed in the document, what the critical concepts sustaining it were, and in what ways its content opened up doors to thinking about sustainability in an unprecedented way.

### 3. Context

*3.1. The Ideology of the Green Deal*

In the 1980s, the EU assumed the position of an expert on environmental policies in the international context (Braun 2013, pp. 26–27). The number of policies and legislation regarding the protection of the environment that go hand-in-hand with rising living standards in the EU ideology has grown exponentially. European sustainability policies have documented environmental discourse of leading European institutions starting with The Single European Act (1987) (ibid.) and ending with the most recent *The European Green Deal* 2020. The EC has maintained this discourse in the *Europe 2020: A European Strategy for Smart, Sustainable and Inclusive Growth* (European Commission 2010) that presented a green-growth plan of self-sufficiency in energy resources, in *EU Biodiversity Strategy to 2020* (European Commission 2011a) that proposed reconstructing ecology as a natural life-enabling asset that has caused economic losses when in crisis and in *Roadmap to a Resource Efficient Europe* (European Commission 2011b) that argued for green growth strategy as a way to overcome the crisis. The discourse and terminology of environmental care and sustainability have been developed extensively by the EC in economic and developmental policies over the last decade. The Juncker Commission of 2014–2019 prioritized geopolitical issues, and security in particular, due to the Ukrainian crisis in 2014 and political tensions with Russia (Siddi 2016). In *An EU Action Plan for the Circular Economy* (2015), the Commission proposed to transition to a different, "sustainable" economy. In *A European Strategy for Plastics in a Circular Economy* 2018, the EC put forward its global leadership in "transitioning to the plastics of the future" and its focus on transforming the EU economy into a "modern, low-carbon, resource and energy-efficient" one. Finally, in the *Green Deal*, the newly elected Von der Leyen Commission pledged to reaffirm its leadership in global environmental challenges.

The objective of the *European Green Deal* is to provide an overall direction for the EU legislation and regional development strategies where climate policy and emissions reduction have become the top priority (Siddi 2016). When praising the *Green Deal*, the EC's President Ursula von der Leyen pointed out its emphasis on making Europe the world's first climate neutral continent by planning for net zero carbon emissions by 2050. In her words, the *Green Deal* is Europe's "new growth strategy" and "man on the moon moment" (Harvey and Rankin 2020). It presents the EU as the incontestable leader with a prominent and well-recognized status in the domain of environmental protection that has made the topic its priority (see EC, *Europe 2020*). Foregrounding the environmental agenda is also a way for the EU to showcase its credibility in the eyes of European and world audience at the time when protecting our environment has in recent years become a key issue shaping the discourse of political leadership. The question that we have pondered is, does the document represent the sort of progress and innovation worthy of the praise and comparison to man's landing on the moon (cf. Slatin 2019)?

The *Green Deal* represents an exceptional chance to promote a particular ideology of environmental protection and sustainability practices, and is supported by the very status of the EU. Ideologies are created by internalizing certain values, maintaining attitudes, responding to social settings, and aiming at social consent (cf. Fairclough 2001; Kress and Hodge 1979). There are no socio-cultural environments, linguistic spaces, texts, and talks without an ideology supporting them and, at the same time, becoming internalized by the population and externalized in behaviors and language. The ideology backing up the *European Green Deal* is built upon the presupposition that Europeans value their natural resources, diversity of natural environment, and social well-being of citizens, and are unified in planning to safeguard these values into the future. However, the *Green Deal* underestimates the citizens' will to act; it calls citizens to cooperate under the leadership of

the EU to protect their values rather than to call for a radical change in habits and resetting of behavioral norms.

The *Green Deal* aims to implement green strategies and goals in the circular economic model, and abandon the linear model. Kirchherr et al. (2017) define circular economy as "an economic system that replaces the "end-of-life" concept with reducing, reusing alternatively, recycling and recovering materials in production/distribution and consumption processes." While the EC documents we examined put the emphasis on recycling, Kirchherr et al. do so on reduction. The EC recognized as compatible environmental protection and GDP growth in its *Europe 2020*, which is considered questionable (see also Ossewaarde and Ossewaarde-Lowtoo 2020 and criticism of growth vs. degrowth). In its *Biodiversity Strategy* 2011, the EC emphasized that the ecological crisis represented economic losses and argued that ecology must be enhanced or rebuilt around the concept of natural assets that make life possible (Ossewaarde and Ossewaarde-Lowtoo 2020, p. 3). The *Green Deal* pledges for the EU to undergo transformation of climate (to become "climate-neutral"), economy (to implement "circular economy"), energy (to depend on "clean energy"), and resources (to become "secondary", i.e., recycled) in order to safeguard the health of its citizens and diversity of the natural environment.

In sum, the text is a political manifesto lodged in the EU environmental discourse and, at the same time, correlated with economic and socio-political values that have been internal to the EU ideology. (Ossewaarde and Ossewaarde-Lowtoo 2020) argue that, "Despite emission reduction standards and the greening of technologies (and hence of production and consumption), the old power structures remain more or less unchanged, legitimized by the green growth discourse . . . " Green growth is an alternative to the so-called "brown growth" that refers to economic development relying on intense fossil fuel consumption without dealing with the impact of this consumption on the environment (World Bank 2013, accessed on 27 December 2020). In contrast, green growth aims at "greening capitalism", thereby shifting the focus on supporting business and industry practices with reduced environmental impact, developing and implementing environmental policy, and technological innovation that would reduce greenhouse gas emissions, all without slowing GDP growth (Ossewaarde and Ossewaarde-Lowtoo 2020, p. 1; see also specific features in which the *Green Deal* transcends the traditional discourse[3]).

### 3.2. Sustainability Criticism: Eco-Innovation and Plastic Waste Generation

Being a prominent, relevant, and highly emotional issue promoted throughout the world, sustainability has also become a channel for political institutions to endorse their presence (see also Blommaert and Bulcaen 2003). But what does sustainability mean? There is no universally agreed upon definition of "sustainability" and the concept continues to be considered "complex, controversial, open-ended and challenging" across scientific and academic communities (Ramos et al. 2020, p. 1). Ramos et al. refer to sustainability and sustainable practices as transformations in "ecological, political, ethical, socio-economic, democratic, cultural and theological dimensions . . . required to avoid crises and a possible future societal and environmental collapse" (p. 2). Engelman (2013) notes that in our age, the word sustainability is used with "a cacophonous profusion" and "can mean anything from environmentally better to cool" (p. 3). Central to the environmental agenda, i.e., "the green norms the EU promotes," are the principles of ecological modernization, sustainable development, and the precautionary principle (Braun 2013, p. 25). The principle of ecological modernization interlinking ecology and economy implies "modernization" of widely employed technologies with the goal to improve efficiency, i.e., to use fewer resources but increase production, as well as environmental safety (see Machin 2019). The principle suggests that the economy benefits from the so-called "eco-innovation" (

---

[3] The authors point out that the EC recognizes the existence and importance of ecosystems, sees the connection between a restored natural environment and human health, points out the problem of traditional industry relying heavily on the extraction of resources, speaks in terms of the transformation of the industrial sector as necessary for climate neutrality but remains vague as to the content of that transformation and remains uncritical towards the ICT sector, pp. 8–9.

Janicke 2008) aimed to minimize negative impact on the environment. The principle of sustainable development adopted by the EU is derived from the *Brundtland Report* of 1987 and represents "development that meets the needs of the present without compromising the ability of future generations to meet their own needs" of economic viability, environmental protection, and social equity (World Commision on Environment and Development 1987). The precautionary principle is defined in the *Rio Declaration on Environment and Development, Article 15* (United Nations, Economic and Social Council 1997) as "lack of full scientific certainty shall not be used as a reason for postponing cost-effective measures to prevent environmental degradation" (par. 80).

Based on the above principles, the EC has defined its development strategies as "sustainable" and made it evident in its public communication. However, as (Braun 2013) pointed out, the EU has been committed to sustainability rhetorically rather than in practice (p. 117; see also Siddi 2016). For one, the above-mentioned principles appear to be conceptual contradictions. For example, the ecological modernization theory, as interpreted by the EU, maintains that a simultaneous economic growth and environmental protection are possible but fails to address the "fundamental contradiction between an economic model fostering unlimited growth rates, and environmental and human resources which are by definition limited" (Lietaert 2008, p. 68). One of the main instruments of ecological modernization is an increase in efficiency, extensively addressed, for example, in the EU *Action Plan for Circular Economy* (2015). However, the increase in efficiency neglects the rebound effect (also known as a Jevons' paradox or a Khazzoom-Brookes' postulate) that states that an efficiency increase paradoxically comes with a higher resource consumption. Thus, even if the objective of the increase is to use fewer natural resources, it is likely to produce the opposite effect. (Braun 2013) writes that the critics of the EU's environmental policy would argue that the EU tends to confuse the sole principle of ecological modernization with sustainable development (p. 117). For three, several problematic areas in implementation of sustainable development policies, and namely plastic waste management, have been scrutinized by organizations such as EEA. Lastly, many of the policies are non-binding and/or take a long time to be adopted by the EU member-states because of the principle of multilateralism, a system of mutual and complex political decision-making the EU has adopted. Moreover, some member-states, most prominently the Czech Republic and Poland, form an informal coalition that opposes strong climate governance (Siddi 2016).

For example, generation of plastic waste has been recognized in environmental and economic policies issued by the EC over the past decade as a major threat to the environment. Since the 2010s, the primary solution to the problem of waste (addressed in the *Action Plan*, European Commission 2015) has been recycling. Nevertheless, the use of plastic in the industry and plastic waste generation rates have continued to increase (European Commission 2018). In 2019, the EEA stated that the solution to the rapid growth of plastic waste in the previous decade has been the so-called plastic waste trade that refers to a complex international trading scheme of sending waste from the EU member-states mainly to Asian countries such as China since the EU supposedly did not have sufficient waste-processing capacities (although growing recycling rates in the region suggest the opposite, according to the association of European plastic manufacturers PlasticsEurope, PlasticsEurope 2018, 2019). The International Solid Waste Association acknowledged in 2014 that almost half of plastic waste collected for recycling in EU member-states was exported abroad and mostly to China (Velis 2014). According to EEA, much of the exported plastic is "mismanaged", i.e., "left uncollected, openly dumped, littered or managed through uncontrolled landfills." As of 2019, the "exports" have shifted to other countries since China had tightened its regulations on waste imports in the last few years. Moreover, Song 2016 pointed out that "garbage recycling can never be 100 percent" due to the entropy principle, the discussion of which tends to be avoided in public domains (p. 17).[4]

---

[4] In thermodynamics, entropy refers to the fact that physical properties of a material can only decrease over time and thus it is impossible to keep a fixed efficiency rate in recycling.

Nozik 1992 criticized the concept of sustainable development on the grounds that it does not stand up to analysis because it, " . . . sanctions the *status quo* of sustainable profit, and cannot, therefore, allow for a radical reform of relationships between people, and between humans and Earth-Nature" (p. 13, quoted in Sauvé 1996).[5] Based on his review of twenty-nine research articles, Ramos et al. 2020 suggested to "rethink sustainability" and "develop new perspectives" and evaluated the current research as being concerned merely with concepts and their definitions, approaches, strategies, policies, practices, roles, and contextual applications of sustainability rather than action. It encompassed "ecological, political, ethical, socio-economic, democratic, cultural and theological dimensions" of the issue but did not advance our thinking on sustainability (p. 2). The research has moved "from an anthropocentric to a more eco-centric or holistic worldview" (p. 2) and from regionally based mechanisms driving environmental practices, individualism, inclusion in global sustainability issues and sustainability enhancement to employment, the role of information technology, artificial intelligence and its tools, learning mechanisms, waste management, behavioral change, and "participatory planning" practices but provided no conclusive stance on the practices or mechanisms that could induce the transformation of the unsustainable into sustainable. Sustainability turned out to be a totalizing concept of universal applicability and an almost exclusive explanatory ability but seized to be useful as a vital concept used as a model of hegemonic development of the 'developed', i.e., industrialized world (Bogiazides 2011). Besides, the production, recycling, export, and reuse of plastics in Europe and Asia document how Europe has been "recycling" by exporting plastic as "secondary raw material". (Bogiazides 2011) pointed out inner contradictions in all the domains sustainability claims to represent, i.e., those of environment, economy, and social equity, noting that it tended to be conflated with viability in economics and to contrast with social equity.[6]

## 4. Results

### 4.1. The Green Deal's Content: Mediating Meaning

Word choices, set phrases, metaphors, and sentential patterns mediate intended messages (Stubbs 2001) and aim at highlighting or marginalizing a topic, emphasizing or sidelining an agenda, as well as gaining trust of the public. Naming concepts central to the topic, relexicalizing familiar words by dressing them up with innovative meanings, using euphemisms, and depending on set phrases are strategies employed to enhance a particular social reality. More than that, the constant rehearsal of patterns and phraseological collocations prioritize and fix meanings in speakers' minds, and establish mind habits through which we tend to think about issues (cf. Stubbs 1997).

Throughout the *Green Deal*, the concept of sustainability is interrelated with that of *energy*, *resources*, *nature*, *climate*, *technology*, *economy*, *power*, *business*, and *social equity*. The major themes identified in Part One are degradation of the environment that is ideologically transformed into a challenge, encouraging a new economic opportunity (relexicalized as *the opportunity for Europe*) that progresses along a *new path* towards changes in populations' behaviors (e.g., *The EU must be at the forefront of coordinating international efforts*; *The policy response must be bold and comprehensive and seek to maximize benefits*). Its developmental strategy is framed by the concept of the environment being transformed into *a globally shared commitment* that aims to engage the public in the process.

Part Two consists of proposals to reduce greenhouse gas emissions and carbon production, renew energy sources, and develop new technologies. The discourse of the proposals

---

[5]   Sauvé (1996) pointed out that the concept of sustainable development is not self-explanatory but subsumes multiple concepts and paradigms, and the many discourses on "environmental education for sustainable development" subscribe to the concept of alternative development. However, the notion of sustainable development cannot adequately express characteristics of alternative development because it is just a particular segment of alternative development.

[6]   Bogiazides (2011) further predicted that, "sustainability [would] continue to hold sway as the all-encompassing imperative in matters of planning and development". Decision makers, at all levels of governance, would continue acknowledging it as "their guiding star, the most potent weapon in their ideological arsenal, enabling them to appear operating for the common good while obfuscating the divisive conflicts of the present", p. 6.

has sustained that of earlier documents. The theme of Part Three is again the EC promoting the EU as the global leader and authority on producing meanings of sustainability that *must be at the forefront of coordinating international efforts towards building a coherent financial system that supports sustainable solutions*. As such, it promises to *continue to promote and implement ambitious environment, climate and energy policies across the world . . . develop a stronger 'green deal diplomacy' focused on convincing and supporting others to take on their share of promoting more sustainable development . . . by setting a credible example* and be *an effective advocate* on climate and environmental measures. In the conclusion, the *Deal launches a new growth strategy for the EU...it supports the transition of the EU to a fair and prosperous society that responds to the challenges posed by climate change and environmental degradation, improving the quality of life of current and future generations*. Both the EU economy and its citizens are to transform their behavior so as to respond to the proposed institutional action taken on the citizens' behalf. Such action fosters hegemony and consequently empowers the EU as the knowledgeable agent, donor, and leader in the field associated with positive actions of *mobilizing industry, tackling emission sources, accelerating transition* or *transforming the economy* that suggest speed, action, and change.

The content is mediated through a special register that identifies technology as the major "enabler" of the proposed or implemented changes endorsing sustainability (cf. [Fairclough 1995](#)). It is applied in the contexts of global economy, climate change, scientific progress, and overall improvement. The technology described as *critical*, *clean*, *innovative*, and *breakthrough* is promised to have a *significant potential* and its primary appeal lies in unifying people around the goals of sustainability (e.g., *The Horizon Europe program will also involve local communities in working towards a more sustainable future, in initiatives that seek to combine societal pull and technology push*). The impersonal nature of the technical register is evident also in its preference for nominalizations (such as *deployment*, *demonstration*, *potential*, *demand*, *frontrunner*, *stakeholder*, or *sustainability*) over simple action verbs with the effect that technologies (along with *programs*, *initiatives*, *the framework*, and *the EU industry*) rather than people drive sustainability and become its agents (e.g., *This framework should foster the deployment of innovative technologies*).[7]

The register works with phrases and terms one tends to associate with business such as *efficiency*, *modernization*, *transformation*, *transition*, *innovation*, and *competitiveness* that have thus become the energizing concepts of sustainability. To render sustainability relevant to Europe's future, the EC has drawn also upon the professional lexicon and phraseology of institutional diplomacy and the domains of economy, technology, and institutional policies, as evident in verbal phrases such as *to boost the market* or *to maximize benefits*, noun phrases such as *strategic plans,* and metaphors influencing certain underlying political and social beliefs and evoking emotional responses ([Simpson et al. 2019](#), 229). The *Green Deal* has innovated its discourse by creating terminology covering new concepts, e.g., *throughput* as in *The EU's industry . . . remains too 'linear,' and dependent on a throughput of new materials extracted, traded and processed into goods*; or *pro-active re-skilling and upskilling*.

### 4.2. Sustainability Phrases and Keywords

Keyword analysis extracts words that are prominent, relative to a point of reference. To assess keywords relevant to the topic of environmental protection and sustainability, we counted manually thirty keywords of the *Green Deal*, considered their context and then chose the twenty that turned out to predominate (cf. [Cohen 2010](#)). As a test, we counted the words (*European*) *Commission* (that appeared 122 times at 1.07% word frequency rate), and *the (European) Green Deal* (that appeared 36 times and has a 0.31% rate). The frequency rate of 0.31% was assumed to be an average word count, and any rate above that a high frequency rate. Out of the twenty keywords, the first eleven had a frequency rate higher than 0.31%,

---

7   There are many additional examples, for instance, *Digital technologies are a critical enabler for attaining the sustainability goals; New technologies and scientific discoveries, combined with increasing public awareness and demand for sustainable food, will benefit all stakeholders; New technologies, sustainable solutions and disruptive innovation are critical to achieve the objectives of the European Green Deal.*

and emerged quantitatively as the crucial identifiers of the text and discourse. The most frequent and repeated words in the *Green Deal* were *climate*, *un/sustainab/ility/ly/le*, *green*, *environment/al/ally*, *energy*, and *emission/s* but also *transition*, *in/action/s*, *invest/ment/ments/ors*, and *econom/y/ies/ic*. The word *green* appeared eighty times (0.7%), mostly in set phrases such as *green economy*, *green agenda*, *green sectors*, *green transition*, *green regulation*, *green budgeting practices*, *greenhouse*, and *green growth*. Very common were also *transition*, *technologies*, and *society* (as in *just transition*, *clean technologies*, and *fair and prosperous society*).

Judging by the word count and currency of certain set phrases, the most important themes of sustainability turned out to be *change*, *transition*, *transformation*, *challenge*, *plan*, *measure*, *regulation*, and *proposal* that endorsed meanings associated not only with environmental protection but also business and economy. To take it a step further, they fixed word patterns, crowded out meanings the reader would expect, and relexicalized their content. In descending order, each word below is followed by the number of text occurrences and their percentage in relation to the overall word count. Given that the total word count of the document is 11.325 words (excluding footnotes and the table at the end), the percentage was calculated by multiplying the number of occurrences by one hundred and dividing the result by the total word count (e.g., 122x100: 11.325 = x%).

| | | |
|---|---|---|
| 1. *climate* | 112 | 0.98% |
| 2. *green* | 85 | 0.75% |
| 3. *(un)sustainab/(ility)(ly)(le)* | 83 | 0.73% |
| 4. *environment/(al)(ally)* | 67 | 0.59% |
| 5. *energy* | 65 | 0.57% |
| 6. *transition* | 48 | 0.42% |
| 7. *invest/ment/ments/ors* | 48 | 0.42% |
| 8. *econom/y/ies/ic* | 47 | 0.41% |
| 9. *in/action/s* | 46 | 0.40% |
| 10. *emission/s* | 44 | 0.38% |
| 11. *develop/ment/mental* | 35 | 0.30% |
| 12. *ensure* | 34 | 0.30% |
| 13. *global/ly* | 33 | 0.29% |
| 14. *financ/e/ing* | 33 | 0.29% |
| 15. *bio/-* | 32 | 0.28% |
| 16. *ambitio/n/us* | 32 | 0.28% |
| 17. *product/s/ion* | 30 | 0.26% |
| 18. *public* | 28 | 0.24% |
| 19. *innovat/ion/ive* | 27 | 0.23% |
| 20. *soci/al//ally/ety/eties* | 22 | 0.23% |

A distinctive topic of the sustainability register in the *Green Deal* is that of economic growth that conceptually stands in the way of sustainability. (Kambites 2014) concurs that the concept of sustainable development has been adapted to conform to the dominant political discourse, emphasizes the incompatibility of economic growth and environmental protection, and hence avoids rather than facilitates radical action. Kopnina (2019) criticized questionable claims such as that, "...profit-oriented production and thus economic growth can be decoupled from natural resource consumption", which in the context of the global economic system is impossible (cited in Jindřichovská et al. 2020, p. 3). The word *growth* being the objective of sustainability strategies appeared eight times in phrases such as *sustainable growth*, *inclusive growth*, *growth opportunities*, and *growth jobs*. To overcome the conceptual obstacle *growth* was relexicalized through phrases such as *new growth strategy for the EU*, described as *sustainable*, *inclusive*, and *decoupled from resource use*. Economic growth was claimed to employ energy dependent on *renewable energy sources*, *secondary raw materials*, and *clean energy supply* to be gained through *clean energy transition*, *sustainable products policy*, *green washing*, and *decarbonization*. Other meanings of keywords were adjusted in similar ways, e.g., *economy* became *clean*, *circular*, and *bio-economy*, to be implemented through *a new circular economy action plan*, *smart integration of renewables*, *smart infrastructure*, and the

*EU industrial strategy.* Economic growth becoming *new growth strategy* is an example of a lexical transformation aimed at seducing readers into imagining new possible ways of dealing with a concept that has already ossified in other contexts. Whether this sort of alternate economic growth is attainable is, however, questionable, in spite of strengthening the practices of ecological modernization. The following examples illustrate the context of economic growth and draw attention to contradictions in meaning:

1, 2. The European Green Deal is a response to these challenges. It is a new growth strategy that aims to transform the EU into a fair and prosperous society, with a modern, resource-efficient, and competitive economy where there are no net emissions of greenhouse gases in 2050 and where economic growth is decoupled from resource use.

3. The EU must be at the forefront of coordinating international efforts towards building a coherent financial system that supports sustainable solutions. This upfront investment is also an opportunity to put Europe firmly on a new path of sustainable and inclusive growth.

4. The European Green Deal will support and accelerate the EU's industry transition to a sustainable model of inclusive growth.

5. Well-designed tax reforms can boost economic growth and resilience to climate shocks and help contribute to a fairer society and to a just transition.

6. The Commission will continue to work on new standards for sustainable growth and use its economic weight to shape international standards that are in line with EU environmental and climate ambitions.

7. To mobilize international investors, the EU will also remain at the forefront of efforts to set up a financial system that supports global sustainable growth.

The register of sustainability is not unique to the *Green Deal*. Its keywords and frequency counts characterize the *Strategy for Plastics* 2018 as well:

| | |
|---|---|
| *environment/(al)(ally)* | 0.51% (cf. the *Green Deal* 0.59%) |
| *circular* | 0.19% (cf. the *Green Deal*, 0.17%) |
| *econom/y/ic/ics* | 0.48% (cf. 0.41%) |
| *global/ly* | 0.26% (cf. 0.29%) |
| *Bio* | 0.21% (cf. 0.28%) |
| *Action* | 0.35% (cf. 0.4%) |
| *invest/ment* | 0.32% (cf. 0.42%) |
| *develop/ment/mental* | 0.51% (cf. 0.3%) |
| *public* | 0.23% (cf. 0.24%) |
| *ensure* | 0.17% (cf. 0.3%) |
| *financ/e/ing* | 0.14% (cf. 0.29%) |
| *innovat/ive/ion* | 0.48% (cf. 0.23%) |
| (un)sustainab/(ility)(ly)(le) | 0.19% (cf. 0.73%) |

Just as the *Green Deal*, *A European Strategy for Plastics in a Circular Economy* 2018 portrays the EU as the "global leader": The EU is taking a leading role in a global dynamic or reaffirms European leadership in global solutions. "Global" typically collocates also with obstacles, action, challenges, opportunities, and effort (e.g., There is a growing awareness of the global nature of these challenges; Opportunities and challenges linked to plastics are increasingly global) and the EC proposals with ambitious. Challenges are promised to be transformed into opportunities to be met with the goal to transform industries, sectors, policies, etc.[8] Economic growth is described to be one of the main objectives, alongside the environmental and societal ones. Although the discourse proposes new approaches, they do not consist merely in developing innovative business models to minimize plastic waste

---

8    For instance, . . . *To move towards that vision, this strategy proposes an ambitious set of EU measures; The Commission facilitated a cross-industry dialogue and now calls on the industries involved to swiftly come forward with an ambitious and concrete set of voluntary commitments*).

at the source, for instance, as a way of achieving not only environmental and social but economic benefits a priori.[9]

### 4.3. LIWC Analysis: Concepts and Word Categories

A function of the linguistic word count (LWIC) software is to display percentages of semantic and syntactic properties of the text and to group words into conceptual categories. It is significant that, according to the LWIC count, among the strongest conceptual magnets of the *Green Deal* turned out to be 'power', 'money', and 'bio'. The Table 1 below presents a summary of prominent concepts and syntactic categories along with words they attracted. The results of LIWC text analysis thus revealed linguistic meanings constructing the concept of sustainability as well as those upon which sustainability has been based.

**Table 1.** LIWC2015 software version Macintosh v.1.6 analysis grouping words into concepts and categories.

| Concept/Syntactic Category | The Most Frequent Words Attracted to the Concepts and Categories |
|---|---|
| 'power' | *confidence, best, lead, leading, leader, ambition, authorities, competence, competitive,* and *workers* |
| 'money' | *trading, banks, business, consumer, economic, finance* |
| 'bio' | *life, heart, food, feed, living, water, healthier, biological* |
| 'drives' | verbs and verb forms: *to protect, to risk, to prevent, to curb, to inhibit, avoided, avoiding, protecting*<br>nouns and words used as nouns: *unwanted, threat, risks, protection, loss, security, prevention, disasters*<br>adjectives *safe, careful, secure, hazardous* |
| 'social' | *communication, together, leader, help, ownership, partners, social, member, encourage, share, group, community, friendly, families* |
| 'compare' | *neither, greatest, lower, lowest, larger, best, higher, stronger, largest, better, healthier, unique, same* |
| Verbs | *warming, passing, being lost, use, must, put, will, bring, accepted, needed*<br>verbs associated with explaining, informing and sharing: *to explain, give, help, inform, meet, send, share, accept, affect*<br>verbs associated with business: *pay, sell, trade* |
| Adjectives | *unique, green, new, fair, modern, natural, same, greatest, active, national, additional, massive* |

### 4.4. Choosing Grammatical Patterns

Structural patterns provide sentential choices delivering thoughts and ideas. A common strategy of transferring the meanings in the *Green Deal* is agent deletion. For instance, a sentence such as *This will be coordinated with action at global level* shows how one can avoid naming who will *coordinate* and *act* (cf. Orwell 1946). Grammatical structures such as the active over passive voice enable identifying agents behind particular actions (e.g., *citizens are fully involved in designing them* [policies]) or withholding their identification (as in *This will be coordinated with action at global level*), and backgrounding or marginalizing targets and recipients.[10] Using the passive verb forms *being lost* and *are being polluted* suggests deliberate ambiguity about who the agents of this loss, pollution, and destruction are (e.g.,

---

9   For instance, *New approaches—developing innovative business models, reverse logistics, or designing for sustainability, for instance, can do much to help minimize plastic waste at source, while achieving further economic, environmental, and social benefits; Stepping up the recycling of plastics can bring significant environmental and economic benefits; Reducing fragmentation and disparities in collection and sorting systems could significantly improve the economics of plastics recycling.*

10  For instance, *The Commission proposal for a Neighbourhood, Development, and International Cooperation Instrument proposes to allocate a target of 25% of its budget to climate-related objectives*, which actually means that *the proposal* rather than EC members *proposes* to allocate rather than *will* allocate 25% to *objectives* rather than specific actions.

*One million of the eight million species on the planet are at risk of being lost. Forests and oceans are being polluted and destroyed*).

On the one hand, the *Green Deal* includes agentive verbs in the form of the collective plural 'we' suggesting inclusion of citizens in actions occurring in their interest but abounds also in personifications and nominalizations that avoid naming actual actors, on the other hand (as in *The biodiversity strategy will identify specific measures to meet these objectives*; *The Commission has already set out a clear vision of how to achieve climate neutrality*; or *This Communication . . . resets the Commission's commitment to tackling climate . . . challenges*). Processes of reducing greenhouse gas emissions and carbon production, renewing energy sources, and developing new technologies are rendered through nominalizations such as *new growth strategy*, *a new path*, *protection and restoration of natural ecosystems*, *consumer protection, the well-being of citizens, climate change*, and *biodiversity* rather than active voice verbs that would indicate who will *grow, protect, restore, renew, modernize, make possible*, or *lead*.[11]

Nominalizations such as *extraction* or *involvement* enable the Commission to distance itself from undesirable situations such as global environmental degradation by omitting the details of who extracts and who is involved (e.g., From 1970 to 2017, the annual global extraction of materials tripled, posing a major global risk; or, The involvement and commitment of the public and of all stakeholders is crucial to the success of the European Green Deal.). Writing that, *The drivers of climate change and biodiversity loss are global* enables not to specify who actually drives climate change. The priorities are to be achieved by means of a *social dialogue, policies, technologies* and *strategies* implementing *skills, energy, resources* and *sources.* Using modal verbs (such as *should, would,* or *must*) enables agent deletion as well (for instance, *There should be both real and virtual spaces for people to express their ideas* where using *should be* is a way to get around identifying who will set up those spaces).[12] Additionally, the commands and imperatives are attenuated so that they become meek suggestions (e.g., *The natural functions of ground and surface water must be restored*). This sort of usage often borders on comprehensibility (e.g., *The Commission will support clean steel breakthrough technologies leading to a zero-carbon steel making process . . .* ) and opens up to ambiguous interpretations.

Many sustainability-related ideas in the document are presented as "targets with indicative deadlines" while others as suggestions, which is related mostly to how the EU has structured its climate energy governance. While greenhouse emission reduction and renewable energy targets are binding for the member-states, efficiency targets are only indicative (Siddi 2016). The *Green Deal* contains many suggestions, cautionary warnings in the form of hypotheticals, and indirect promises (e.g., *If the risk materializes, there will be no reduction in global emissions; The commission has been working to provide Member States with new financial resources; The EU will put emphasis on supporting its immediate* neighbors; or *The EU will establish innovative forms of engagement*). The *Deal* aims to propose and evaluate, which it does with caution (e.g., *These policy reforms will help to ensure effective carbon pricing . . . ; The Commission will consider measures to improve the energy efficiency . . . ; A combination of measures should address emissions . . .* ). The measures against the international waste trade are addressed merely as a recommendation: *The Commission is of the view that the EU should stop exporting its waste outside of the EU and will therefore revisit the rules on waste shipments and illegal exports.* In the *Strategy for Plastics*, the EC avoids naming the agency when describing the process of dealing with plastics (*a significant share* [of plastics] *leaves the EU to be treated in third countries where different environmental standards may apply*) and refers to it as

---

11  Nominalization is a language strategy of avoiding active verbs along with their agents, and replacing them with deverbal nouns.

12  Similarly, in the sentence *Participants would be encouraged to commit to specific climate action goals* the modal *would be* avoids indicating who will encourage the participants.

*treatment, management, or processing* when suggesting global solutions.[13] Both the *Green Deal* and the *Strategy for Plastics* sideline issues that are critical to resolving the environmental endangerment. Although growing waste generation, insufficient recycling capacities and the international waste trade have been identified as major obstacles to environmental protection by the epistemic community, they remain marginal in both documents.

### 4.5. Summary

The linguistic analysis makes it evident that the EC has constructed a sustainability discourse and employed particular keywords, set phrases, and grammatical patterns to endorse its traditional agenda and prioritize familiar topics. The *Green Deal* does not depart from them in any major direction. As a result, sustainability remains a key topic to be talked up in Europe by the institutions of power that mediate knowledge and seek to position readers to support it. By doing so, they foster their hegemony and control the public discourse of sustainability (cf. Buckledee 2018) and, by implication, also consumers of the discourse.

We suggest that technical difficulty of the lexicon, lack of transparency in ways that meanings are mediated, and ambiguity in the identification of agents and actions suggest institutional ownership of the discourse and reluctant sharing of power that is invested in it. The technical and impersonal nature of the sustainability register makes it difficult for readers unaccustomed to the institutional jargon to follow the text (see also van Dijk 2008; Drew and Heritage 1991). Meanings behind words and set phrases are mediated in a way that does not empower readers themselves but the EC as the expert (cf. Orwell 1946). That "discussion of the topic melts into the abstract diction" is evident in statements throughout the document, such as *There is significant potential in global markets for low-emission technologies, sustainable products and services*.[14] At times, the meaning becomes submerged and disguised in sentences that border on comprehensibility (e.g., *The strategic plans will need to reflect an increased level of ambition to reduce significantly the use of chemical pesticides*) (see Simpson et al. 2019). Instead of providing access to the latest information on environmental protection, the *Green Deal* jargon, being technical and heavily institutionalized, tends to disguise meanings of everyday words assumed to be self-evident and readily comprehensible by relexicalizing them (see Agar 1985).[15] As a result, the very words create a communication barrier instead of a communication channel. In our assessment, the set phrases, neologisms, and word collocations associated with sustainability, climate, and environmental protection have prevented an open dialogue across generations and professional backgrounds. Among others, they stand in the way of "talking" with the young vocal activists (see also Rydin 1999).

The *Green Deal* boasts a significant rhetoric power but its jargon is far from being transparent in meaning, which weakens the actual content. Its rhetoric seems to fail those whom it should impact, i.e., the citizens; it demands very little when it comes to their inclusion and puts forth no significant expectations of social transformation (cf. Ossewaarde and Ossewaarde-Lowtoo 2020). Thus, we question whether modernization and technology, green or plain, can substitute for social determination that could cause a radical shift in behaviors of citizens and communities. As Fischer and Forester (1993) write, "policy-making is a discursive struggle" (1993: 1–2). To analyze this struggle critically is to inspect how the discourse has been communicated to the public, whether it was made accessible to the concerned population, how it formulated the basic arguments,

---

[13]　These solutions are evident in the following excerpt: *The Commission will promote the development of international standards to boost industry confidence in the quality of recyclable or recycled plastics. It will also be important to ensure that any plastics sent abroad for recycling are handled and processed under conditions similar to those applicable in the EU under rules on waste shipments, supporting action on waste management under the Basel Convention, and developing an EU certification scheme for recycling plants.*

[14]　Abstract diction saturates claims throughout the *Green Deal*, e.g., in the statement, *"To keep its competitive advantage in clean technologies, the EU needs to increase significantly the large-scale deployment and demonstration of new technologies across sectors and across the single market."*

[15]　Relexicalization is a strategy of modifying semantic meaning habitually associated with a lexical item and endorsing an alternative one supporting an ideology that is prioritized.

what language strategies of communication and information dissemination it employed, what ideology the discourse enhanced and what agenda it could have hidden (see also Rydin 1999). All these questions concern policy-making in the *Green Deal.* We suggest that sustainability policies have become the tool of political negotiations because the very concept of sustainability is ambiguous, which can be an advantage, allowing those with different, conflicting interests to "reach some common ground upon which concrete policies can be developed" (1993: 1–2).

A prosperous economy that is decoupled from energy resources is an appealing concept implying that economic growth is a "sustainable goal", according to (Ward et al. 2016). However, a consensus among economy scholars is that decoupling economic growth and resource use is not only unlikely but to develop growth-oriented policy around that expectation is misleading; social prosperity needs measurements that are more intricate than economic objectives such as GDP alone (Ward et al. 2016). (Ossewaarde and Ossewaarde-Lowtoo 2020) concur that the *Green Deal* devalued a new social pact by involving mostly strict calculation. We agree that the *Green Deal* has missed the opportunity to call for an imminent social change.

## 5. Discussion

### 5.1. Circulating the Discourse and Creating Hegemony of the EU Public

(van Dijk 2008) emphasized the fact that discourse of powerful institutions and organizations is the essential social practice that mediates and manages beliefs of members of a particular society. The EC discourse does exactly that. It is controlled by symbolic elites such as ministers, heads of state, and high-status politicians, and authored by experts who "exercise power on the basis of symbolic capital, . . . shape the discourse [and are] the manufacturers of public knowledge" (p. 35). The strategies to manufacture public opinion consist of supplying institutional information to press releases, press conferences, interviews, leaks, or other forms of preferred access to news-makers" (2008, p. 36; see also (Bourdieu 1991). In this manner, the meanings circulate, become established and fix our thinking on the issue. The sustainability register has also been circulating in reports of domestic and international media that support both the discourse and the EU's role in it. By reproducing the perspective of an uncontested support of the EU policies, the media inspire conversations of sustainability. Cumulative effect of the media presence, i.e., the ways they reproduce decisions and actions as well as their corporate embedding guarantees their influence in promoting the EU as the authority on sustainability empowered to set the environmental agenda.

In European media outlets as well as international public discursive spaces, it is not at all uncommon to see the EU being championed for its "ambitiousness" regarding the sustainability efforts (Harvey and Rankin 2020). Mainstream international news channels call the "green programs" of European politicians "ultra-progressive" (O'Sullivan 2020). *The Guardian* quoted the EC President Ursula von der Leyen that, "The Green Deal is, Europe's man on the moon moment [and] . . . our new growth strategy" described as proposing a change that is ambitious, transformative, disruptive, and unprecedented since, " . . . Nothing similar has been attempted before since previous attempts were piecemeal, limited in scope and sometimes flaccid in execution". The *Green Deal* policy is supposed "to improve people's quality of life through cleaner air and water, better health and a thriving natural world . . . without reducing prosperity" (Harvey and Rankin 2020). As an advantage of the *Green Deal* the authors further highlighted low-carbon economy, mandated renewable energy, reduced air pollution, focus on halted species loss, reduced waste, and better use of natural resources to be achieved through "clear overarching targets" while "efficiencies in resource use would repay the cost of these changes." These advantages were contrasted with "relentless [environmental] exploitation" that allegedly took place before the *Green Deal* was put in action. More than that, the changes catered to citizens and businesses by supplying them with "jobs ... in new high-tech industries from renewable energy to electric vehicle manufacturing and sustainable building" (Harvey and Rankin 2020). It is

worth noting that Harvey and Rankin attributed developments associated with the *Green Deal* to a universal "pattern of human progress" happening since the industrial revolution as if the progress was determined a priori by the way that the *Green Deal* projected it (cf. Bogiazides 2011 criticism of claims of sustainability's universal relevance).

The discourse of sustainability has been followed by European institutions such as national governments, municipalities, and think-tanks seeking to employ "sustainable" practices and endorse changes assumed to be progressive and beneficial for the environment. For instance, several European cities developed a worldwide reputation for promoting "ecological thinking" as a part of sustainable development, which is also reflected in agendas of the EU member-states regarding sustainability, environmental protection, or climate governance. One of them has been the Finnish capital that adopted language usage that can be described as positive, proactive, and based in sustainability-associated phraseology in its public communication website *My Helsinki* (www.myhelsinki.fi/en). *My Helsinki* web includes not only the sections *See & Do, Eat & Drink, Work & Study, Business & Invest,* and *Info* but also *Think Sustainably* providing city visitors with information on *Small carbon footprint activities*, *Responsible and long-lasting shopping*, and *Sustainable treats*. In *Work & Study*, we read that, "Helsinki is a model city of sustainable development . . . [where] ecological thinking is strongly present in everyday life . . . the city places great emphasis on encouraging environment-friendly lifestyles . . . pursues a decrease in emissions, aims for carbon neutrality, encourages to make ecological everyday choices, [takes] climate-related actions, designs tools to track the progress, and aims to reduce lavish over-consumption." *My Helsinki* proudly displays sustainability objectives and goals that are "more ambitious than in the rest of the EU region."

The sustainability register has been reaching the public also by circulating in articles on "green innovations" in major European cities that frequently appear in magazines such as *Forbes*, *Dezeen*, or *Citylab* and describe sustainable urban environments as "smart". For instance, in the feature on London's "pollution-absorbing bench," "smart" urban solutions to pollution celebrate the apparent fact that the bench can "absorb as much pollution as a small forest providing smart measurement of the surrounding environment" (Nace 2018). Not only is this bench described as "practically self-sustainable and self-sufficient" but it can also "display advertising" and appeal to financial investment, as emphasized. Similarly, the design magazine *Dezeen* with an international renomé published an article featuring "the smartest neighborhood in the world" in the Brainport Smart District in the city of Brandevoort in the Netherlands (Carlson and Tatari 2020). Its designer Mariarthi Tatari, the leading architect of the firm described the district as "human-centered" and "smart" since supposedly allowing its citizens to use data-driven methods such as an analysis of information collected from citizens' smartphones so that they provide and receive information inside the district in order to create a "more human environment." In other words, the creators of the district were praised for aiming to employ technology in a more efficient way to save resources. Resource saving by means of technology was presented as a fact no matter how questionable. The district would supposedly, " . . . use sustainable materials [in its construction and maintenance] if possible" and "circular economy in construction, energy, water, food and care." In that way, everyone could participate in maintaining the district, enjoying a "clean and green environment" and "embrace[ing] sustainability" through "new energy systems." Such a representation may be appealing but it is certainly misleading.

Phraseology of the register recurred also in Bloomberg-owned *Citylab*, an online magazine on urbanism, architecture, and design focused on "progressive" urban development. In the article *In Paris, a very progressive agenda is going mainstream*, the author praised the agenda of Mayor Anne Hidalgo in which "sustainability wins" (*Citylab* 2020). What is described as progressive in relation to Hidalgo's political agenda are "forthright, broadly pro-green choices" supported by most city officials agreeing that in Paris, all municipal officials or candidates "strive to green their program." Initiatives described as sustainable include, among others, the so-called participatory neighborhood planning, affordable hous-

ing, or help with the renovation of housing that city landlords struggle to rent; these are, essentially, proposals for a more "efficient" use of urban spaces. That *Citylab* designated the above issues as central to the governance of Paris demonstrates impact of the sustainability discourse as well as the EC's "smart" use of the media space to spike its image.[16]

Green thinking' has become a substantial part of modern European identity that is rooted historically (Strasser 2013) and is also evident in present-day practices that follow the *Green Deal* discourse in believing that the ecological crisis can be transcended through green technologies without compromising prosperity and standards of Western lifestyle (see also Ossewaarde and Ossewaarde-Lowtoo 2020, 6). The results of a recent Eurobarometer poll evaluating citizens' opinions on the environment-related issues indicated agreement and a "general satisfaction" with the environmental efforts carried out by official institutions, and showed that the vast majority of respondents was "personally committed" to protecting the environment (Eurobarometer 2017).[17]

*5.2. The Parallel Discourse of Sustainability: Challenging "Growth"*

The EU developed a discourse on sustainability that has been broadly disseminated but not correlated systematically with the discourse of environmental sciences. Despite the persistence of the official discourse on sustainability, alternative voices critical of the strategies, and objectives in the EU's sustainable development have existed all along. For example, the European Environmental Bureau (EEB) and EEA were endowed with the mission to preserve Europe's environment, enable citizens to address environmental issues, and provide criticism and guidance to the EU regarding environmental policy.

In 2019, EEB issued a public communication report on accessibility of environmental information pointing out that this information was hard to access for the very citizens who were supposed to be its recipients participating in the goals and strategies. The report indicated that although there were various regulations on both the EU and member-state level requiring "active and systemic dissemination of environmental information to the public" (European Environmental Bureau 2019, p. 4), several issues prevented adequate functioning of these regulations. The first issue the report investigated was information accessibility provided to the member-states. Another complaint raised was that environmental information might have been publicly accessible but was not presented in a user-friendly way, making it hard for the citizens to become engaged and environmentally aware. In many instances, the information was supposedly "incomplete," "unreliable", and "outdated" (European Environmental Bureau 2019, p. 5). A poor structure of online informational sources made it difficult for citizens to fully understand relevant environmental issues. The report identified cases when citizens were not informed about major environmental decisions in their region and emphasized that, " . . . Informing the public about emissions and pollution is essential as it is information that can affect their health, wellbeing, and is part of their 'right to know'" (European Environmental Bureau 2019, 9). The report called for the government and local authorities, in particular, to become fully responsible and enable communities and citizens to share environmental information in order to "harness the energy of the public in gathering environmental data." The report emphasized the responsibility of public authorities in providing adequate environmental information, and facilitating and promoting the development of citizen initiatives (European Environmental Bureau 2019, p. 12). (Ossewaarde and Ossewaarde-Lowtoo 2020), praised the *Green Deal*

---

[16] The discourse has spread in academia as well is the article *Environmental Action Programmes of the European Union: Programmes Supporting the Sustainable Development Strategy of the European Union* describing the EU environmental policies projected until 2020 as programmatic and the EU as having an active role in producing meanings of sustainable development (Halmaghi 2016, Romanian military education institute).

[17] The data came from an officially commissioned source and show that the EU discourse has echoed throughout ordinary citizens' opinions displayed in social media. On the start of the current coronavirus pandemic the following post of a long-term resident of Europe appeared in social media confirming currency of the image that the EU has sought to establish for itself, i.e., being an authority on sustainability and environmental protection: *Hold on, my beloved Europe. The European Union consists of the most noble, intelligent and educated people. Europe is the largest donor of humanitarian aid and the largest donor of clean air for the whole world. Here, we have the most radical measures to save our planet. [Now,] we have to pay for the mistakes, dirt and negligence of others.*

for going a step further when calling for a consumer policy enabling informed consumer choices, disclosing fully not-so-sustainable production processes, products themselves and their disposal, and expecting a similar disclosure from companies and financial institutions about their "climate and environmental data" (pp. 8–9).

European Greens (2015) (representing a political party in the European Parliament) expressed concern and skepticism regarding sustainable development policies as well. In a recently-issued brief, they noted that, "The circular economy on its own can improve the availability of materials but cannot provide an endless and increasing supply." The Greens stressed the necessity to face up to the limits of our resources and work towards an absolute reduction of resource extraction. In addition, they criticized the policies on waste noting that recycling was the last resort rather than a solution. Calling attention to the EU's discourse on sustainability, EEB reflected on economic policies of the EU and issued a report evaluating principles of sustainable economy entitled *A circular economy within ecological limits* (European Environmental Bureau 2020). The main objective of "circular economy" was identified as "reducing the absolute quantity of natural resources that enter our economy, and reducing the quantity of waste coming out." The two principles of economic measurements were prioritized as "setting a headline target to halve EU material footprint by 2030" and "setting a cap on absolute waste generation per capita", and thus reducing the intensity of resource input practice. In contrast to the *Strategy for Plastics* and the *Green Deal*, *A circular economy* described solutions proposed by the EC as "one-dimensional" and reflected distrust and dissatisfaction with the EU implemented measures. The report identified current EU efforts as unsatisfiable, noting that "multiple indicators suggest Europe is not moving fast enough in the right direction" and stressing problems such as high amounts of waste generation, concerns related to "green labeling," low levels of recycled and reused materials, and ineffective regulatory measures.

Other critical sources of the official EC discourse directed public attention to lobbying practices in the major EU structures that involved special interest groups hiring professional advocates to argue for specific legislation at the official level of decision-making and thus influenced environmental decisions. Such practices are legally allowed in the EU but regulating them differs at the national level.[18] For instance, the 2019 joint report by the Corporate Europe Observatory, Food & Water Europe, Friends of the Earth Europe, and Greenpeace criticized lobbying practices regarding environmental and sustainability policies of the major resource intensive corporations in the European Parliament and the European Commission, noting that, "Since 2010, just five oil and gas corporations and their fossil fuel lobby groups have spent at least a quarter of a billion euros buying influence at the heart of European decision-making". Greenpeace, known for its confrontational stance and upfront language, further stated that, "It's part of a decades-long strategy by fossil fuel lobbyists of denying widely accepted science, and trying to delay, weaken, and sabotage climate action–despite knowing their business heats the planet and destroys communities."

Siddi (2016) argued that climate governance (and consequently the effectiveness of the *Green Deal* in terms of meaningful changes in the nearest future) relies too heavily on "large private financiers," many of whom are involved in fossil fuel-intensive industries and "unlikely to prioritize long-term climate considerations over short-term profit" (p. 9). While many EU policies are designed to effectively deal with environmental issues, solutions adopted in the actual decision-making are weaker than they could be. For example, in 2014 when the EU agreed on its 2030 climate and energy targets, the five biggest oil and gas companies (Shell, ExxonMobil, Chevron, Total, and BP) declared spending 34.3 million EUR on influencing the officials; between 2014 and 2019, there were more than one meeting a week (and 327 in total) between lobbyists of these corporations and the Juncker Commission. According to the report, the proposed targets were "far from what is required" to keep the atmosphere temperatures from rising above the acceptable levels. The lobbying has also happened at the national level when companies that directly depended on fossil

---

18  For specific national regulations, see lobbyeurope.org/rules-and-regulations/.

fuel extraction sponsored major climate talks such as the United Nations Framework Convention on Climate Change, or threatened to sue the state if it came to adopting ambitious legislation such as a ban on renewal of permits to extract fossil fuels (Greenpeace European Unit 2019).

Criticism regarding environmental impact of current economic practices came also from radical political and academic domains. For example, Michael Lowy, a French-Brazilian Marxist sociologist, philosopher, and author of *Ecosocialism* argued for an aggressive pursuit of environmental preservation to avoid destruction of systems considered unjust by the so-called socialist ecology (Lowy 2015). He stressed that limits on growth and, foremost, the patterns of consumption were essential for sustainability. The concept of "degrowth" becoming a cultural, political, and social movement, and an academic discipline first appeared in France in the 1970s with an objective to address the "dynamics" of infinite growth and relentless pursuit of the economic agenda (Lowy 2015). Degrowth tackles the very belief that growth, especially economic, is a necessary goal for fair and sustainable societies (Demaria et al. 2013; Whitehead 2013). Whitehead (2013) noted that degrowth was, in fact, a "sister concept" of sustainability and an effort to imagine a "downsized world" where the main objectives were voluntary simplicity regarding material possession, a focus on gaining deeper insight on social justice and human wellbeing in general, and a heightened attention to the preservation of natural environment (p. 142). Over the last decades, such objectives were present in grassroots social movements such as zero-waste or cohousing. The defining moment for degrowth was *The First International Conference on Economic De-Growth for Ecological Sustainability and Social Equity* organized in 2008 in France. The panel board consisted of scientists, academics, activists, educators, and social workers who actually argued against some of the central objectives that were later adopted by the official EU discourse on sustainable development. (Ossewaarde and Ossewaarde-Lowtoo 2020) pointed out that in contrast to growth and green growth that leave established oligarchical power structures intact, degrowth is about cultural transformation that consists in negating ecologically irresponsible middle-class lifestyles and corresponding mythologies (p. 2).

As this alternative discourse indicates, sustainability has also become an attempt to reimagine societal values so that focused on the global nature of environmental issues. In this discursive space, a transformational path towards a more sustainable society is often linked to inequality due to the disproportionate distribution of resources among socio-economic groups. Global inequality has thus become an obstacle on the path towards sustainability. As emphasized by Whitehead (2013), "From a humanistic point of view, populations living today in high poverty conditions, a de-growth process cannot be imposed or even presented as a necessary condition. In fact, de-growth patterns have to be designed in function of the impact communities/countries have on the degradation of the planet" (p. 144).[19]

## 6. Conclusions

### 6.1. Practical Limitations and Implications for Further Research

Having scrutinized sustainability discourse of the EU, as represented in the *European Green Deal*, limitations placed upon practical implications of our research stem from the fact that critical discourse analysis can reveal problems and generate discussion about them but cannot solve them. Nevertheless, informed criticism should impact the EC's politics, policies, as well as practices. However, we consider it unlikely for the EU to replace its sustainability discourse or modify its green growth strategies in the near future, unless large-scale unprecedented environmental disasters take place in Europe because

---

[19] (Ossewaarde and Ossewaarde-Lowtoo 2020) remind us that, ... "The failure to transcend the frame of mind that corresponds to that system results in the neglect of community-based solutions, that is, of local democracies that determine the direction and content of their local economies, including the amount of energy used. Energy commons could have been mentioned and highlighted. From the standpoint of inclusion and justice, the inclusion of citizens in political decision-making regarding their earth system must arguably be more than getting them to cooperate with authorities, industry, and the EU's administrative bodies", p. 10.

the symbolic power of capitalist societies has been ultimately dependent on accumulation of material resources: "Especially in late capitalist societies, sustained economic growth has come to depend upon the growing administrative rationality of the state—including especially its administration of the economy—and the legitimation of the state, in turn, has come to depend upon its capacity to sustain capital accumulation within its territory" (Ashley 1988, p. 247). Ashley 1988 argued that states (as well as the EU since it represents an unusually cohesive coalition of states) engage in "heroic practices" of creating a dichotomy between the safety inside the state and the danger of "international anarchy" since there is no supreme international authority presiding over them. Controlling this order involves the use of "hard" tools such as an increasing military presence and monetary resources, and including production and accumulation of material goods. Radical changes such as degrowth strategies contradict the tools.

Our paper is focused on providing research implications (outlined below) rather than managerial and policy implications. We suggest to accept the challenge questioned by environmental and social justice scholars as the target of follow-up research: What will be an effective catalyst of transformational change regarding the understanding of sustainability and its practices in the European Union? This question has remained unanswered by scholars across the disciplines and is likely to dominate debates on sustainable development in European cultural and social domains for years to come.

### 6.2. Overall Conclusions

The official discourse on sustainability, as exemplified in the *Green Deal*, has been built upon the registers of economics, business, and environmental science, and attests to a paradigm shift from unsustainable society to a society that is sustainable. The discourse continues to hold power in the EU, shape the EU's image as a "green leader", and influence how the rest of the world understands sustainability. However, if the discourse is to prevail into the future, it will lead indirectly to additional environmental degradation and cause further social and economic inequality among people and communities because it does not oppose practices that are defined as unsustainable by the epistemic community. The practice of "managing" plastic waste, in particular, questions the soundness of official strategies of sustainability and economic development hidden in the discourse agenda of economic growth. The European economy may be driven by economic growth and maximizing profits through material production but operates, nevertheless, in the context of finite natural resources. Although the EC suggests that it is possible to decouple economic growth from natural resource use, it was proven impossible by global and local economic systems.

The parallel discourse of sustainable development created by the scholarly community and environmental organizations primarily within the ideological space of degrowth exists in the EU, nevertheless. In this discourse, sustainability principles have been shaped by minimizing the human impact on the environment, reducing the economic activity by dematerializing the economy, and ensuring social justice and equality. The parallel discourse does not represent the mainstream of sustainable thinking in Europe, despite its being backed up by the epistemological community. Some of the challenges this community faces are the corporate and industrial interests blocking efficient legislative changes that could alter Europeans' socio-economic values, understanding the social impact of progress and development, and of the present relationship between humans and nature.

It cannot be denied that the ongoing pandemic has both mobilized the societal resources and dominated the spaces of public discourse. Despite this epidemiological crisis, a meaningful change towards sustainability as defined by the actors opposed to the mainstream understanding of sustainability and sustainable development (such as an absolute waste reduction or unjust practices) will depend on reshaping the understanding of sustainability in both European and global contexts, to which further research on critical discourse analysis should pay close attention. Words may be abstract in terms of representing reality, arbitrary in ways they relate to meaning and symbolic of what they represent. But

they matter. We speculate that if the current discourse retains its status quo, it will cause intensification of climate change and environmental degradation.

**Author Contributions:** Both authors contributed to the general research, data analysis and assessment of sustainability per se. E.E. was in charge of the linguistic analysis. O.K. was in charge of the theoretical framework of sustainability. All authors have read and agreed to the published version of the manuscript.

**Funding:** This research was partially funded by Metropolitan University Prague grant number E64-82.

**Conflicts of Interest:** The authors declare no conflict of interest.

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
