# Peer review of "Sustainability in the European Union: Analyzing the Discourse of the European Green Deal"

_jrfm, doi:10.3390/jrfm14020080_

Round 1
Reviewer 1 Report
I think, honestly, that this article is exceptionally good. Whilst my understanding of the methodological elements of the article are not acute on account of a different specialism, it is understandable (which is very important, as I am sure you will agree). I think it is a fascinating angle from which to assess the sustainability agenda of the EU. I wonder whether the examples of not concentrating enough on things like waste management are enough to show a lack of development from the EU, or whether it is just a case that, in reality, they cannot focus on everything all at once? However, your article does position it well enough to demonstrate that it is that much of an issue that it should be considered, as an example. I was fascinated by your inclusion right at the very end about special interests blocking development in this area, and wonder whether you could focus on that a little more in the article, although it is merely a suggestion. I think whilst it may be seen as something connected but separate to your analysis, it is likely closer than the article suggests (if that makes sense). However, that is a judgment call rather than something that should be included.
Overall I am particularly positive about this article and congratulate you for it - so much so that I have asked the Editor to inform me if/when it is published as it would be my pleasure to incorporate its findings into a book I am currently writing.
Author Response
Dear Reviewer,
Please find our cover letter attached.
Eva Eckert and Oleksandra Kovalevska

Reviewer 2 Report
Please, provide more explanation and argument for justifying the importance of the objective of the paper. Why is so important to analize this discourse and not others?
Are there any other similar works? Which are their contribution?
Please provide more connection of the terms with impotant papers that defines them: sustainability, green growth, degrowt, circular economy.......and so.
Implications for research and practioners are neccesary.
Are there any limitations in the paper?
Author Response
Dear Reviewer,
In response to the reviewers’ suggestions, we have revised our paper and are ready to resubmit it to your journal. Below, we explain our revisions, step-by-step.
- We’ve defined the key concepts, i.e., circular economy, green growth, degrowth, and sustainability, in reference to scholarly publications (par. 2 on p. 4; par. 3 on p. 4; par. 2 on p. 20, respectively).
- We’ve justified the importance of the objective of the paper and preference given to the critical analysis of the Green Deal We’ve further explained the merits of discourse analysis, and answered the question about contribution of similar works.
3) We’ve thoroughly revised the Introduction to incorporate and further discuss the concepts defining sustainability with reference to relevant scholarly sources (previously included in chapters 3 to 6). Introductory paragraphs address critical documents relevant to the Green Deal now and contain references to additional published analyses of the Green Deal. In particular, we’ve drawn on Ossewaarde and Ossewaarde-Lowtoo 2020, as recommended. 4) We’ve incorporated the LWIC software as an additional tool to support our linguistic analysis of the sustainability discourse and explained the necessity of conducting the analysis and the methodology in more detail. 5) We provided an explicit presentation of our Results and explained how they supported our Conclusions. |
- We’ve reflected upon implications for research.
In summary, the major changes were (1) reorganization of the materials according to the IMRAD format; (2) elaboration of our methodology by incorporating the LIWC2015 word-counting linguistic software to substantiate our data quantitatively and support our results; (3) explanation of why necessary to conduct critical discourse analysis of the European Green Deal; (4) definition of key concepts and reflection upon documents affected by the EC discourse of sustainability; (5) introduction of outcomes published in a recent article on the topic of the European Green Deal discourse; and (6) addressing limitations of the paper (par. 2 ff. on p. 21), as well as implications and recommendations for further research (the final paragraphs).
We are grateful to the reviewers for their valuable input.
Eva Eckert and Sasha Kovalevska

Reviewer 3 Report
I challenge authors to revise the structure of the paper. The IMRAD structure would suit the paper much better. The introduction section should contain more information about the concepts of sustainability and Green Deal from chapters 3-6. The Methodology chapter should be followed by chapter Results with subchapters 7-8. The rest of the paper should be in Discussion & Conclusion.
Some relevant documents regarding the Green Deal should be cited in the first paragraph of introduction.
The introduction should contain references to other published analyses of the Green Deal with pointing out their results such as https://www.mdpi.com/2071-1050/12/23/9825
Author Response

(The authors gave the same response as above.)
